# WEAK-SIGREG: COVARIANCE REGULARIZATION FOR STABLE DEEP LEARNING

**Habibullah Akbar**
Kreasof AI
Jakarta, Indonesia
`habibullah.akbar@kreasof.my.id`

## ABSTRACT

Modern neural network optimization relies heavily on architectural priors—such as Batch Normalization and Residual connections—to stabilize training dynamics. Without these, or in low-data regimes with aggressive augmentation, low-bias architectures like Vision Transformers (ViTs) often suffer from optimization collapse. This work adopts Sketched Isotropic Gaussian Regularization (SIGReg), recently introduced in the LeJEPA self-supervised framework, and repurposes it as a general optimization stabilizer for supervised learning. While the original formulation targets the full characteristic function, a computationally efficient variant is derived, **Weak-SIGReg**, which targets the covariance matrix via random sketching. Inspired by interacting particle systems, representation collapse is viewed as stochastic drift; SIGReg constrains the representation density towards an isotropic Gaussian, mitigating this drift. Empirically, SIGReg recovers the training of a ViT on CIFAR-100 from a collapsed 20.73% to 72.02% accuracy without architectural hacks and significantly improves the convergence of deep vanilla MLPs trained with pure SGD. Code is available at github.com/kreasof-ai/sigreg.

## 1 INTRODUCTION

The success of Deep Learning is often attributed to the interplay between over-parameterization and specific architectural choices that smooth the optimization landscape (1; 2). However, when these architectural safeguards (e.g. batch normalization, residual connections) are removed, or when architectures with low inductive bias like Vision Transformers (ViT) are trained on small datasets with heavy augmentation (3), optimization often becomes unstable or collapses entirely (4).

This problem is approached through the lens of distributional stability. Intuitively, the evolution of hidden layer representations during training can be likened to a system of particles evolving under stochastic dynamics (Dean-Kawasaki dynamics) (5; 6). In this view, the "stochastic flux"—noise introduced by finite batch sizes, high learning rates, and augmentations—can cause the representation density to drift into degenerate states (dimensional collapse).

To counteract this, **Sketched Isotropic Gaussian Regularization (SIGReg)** is leveraged. Originally introduced by Balestriero & LeCun (7) for the LeJEPA self-supervised learning (SSL) framework, it is adapted here as a plug-and-play loss term for supervised optimization. The contributions of this work are:

1. **Supervised Stabilization:** It is demonstrated that SIGReg is not just an SSL tool, but a fundamental stabilizer that fixes optimization collapse in ViTs trained with AdamW.

2. **Weak-SIGReg:** A simplified formulation is introduced that enforces covariance isotropy via random sketching (8), offering similar stability to the original (Strong) SIGReg with reduced computational overhead.

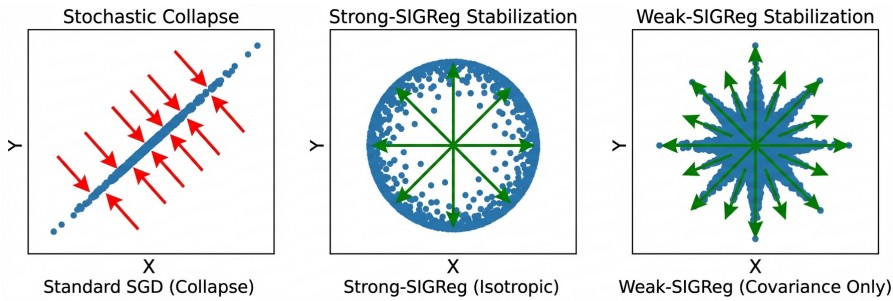

Figure 1: **Visualizing Optimization Dynamics.** (Left) **Stochastic Collapse**: In standard SGD without normalization, representations (blue dots) tend to collapse into low-dimensional manifolds (red arrows). (Center) **Strong-SIGReg**: Forces the distribution towards a perfect isotropic sphere. (Right) **Weak-SIGReg (Ours)**: Targets only the covariance, preventing collapse (green arrows) while allowing more geometric flexibility (star shape), sufficient for supervised stability.

## 2 METHOD: FROM STRONG TO WEAK SIGREG

Consider a neural network encoder $f_\theta(\cdot)$ producing embeddings $Z \in \mathbb{R}^{N \times C}$ for a batch of size $N$. The proposed goal is to regularize $Z$ such that its empirical distribution approximates an isotropic Gaussian $\mathcal{N}(0, I)$.

### 2.1 STRONG SIGREG (LEJEPA FORMULATION)

The original formulation, which denoted as **Strong SIGReg**, was introduced in LeJEPA (7). It minimizes the distance between the Empirical Characteristic Function (ECF) of the embeddings and the analytical CF of a Gaussian. To overcome the curse of dimensionality, it utilizes a random projection matrix $A \in \mathbb{R}^{C \times K}$ to project embeddings into a lower-dimensional sketch space $K$. By matching the CF, Strong SIGReg theoretically constrains all moments of the distribution.

### 2.2 WEAK SIGREG (PROPOSED FORMULATION)

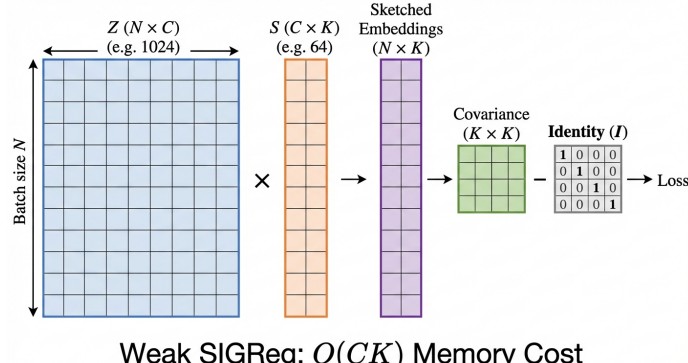

Figure 2: **Weak SIGReg Architecture.** A high-dimensional batch $Z$ ($N \times C$) is projected via a random sketch matrix $S$ ($C \times K$) into a lower-dimensional embedding. The covariance of this sketch is computed and forced towards the Identity matrix $I$. This avoids computing the generic $C \times C$ covariance, reducing memory cost to $O(CK)$.

While matching all moments is theoretically optimal, it is hypothesized that preventing dimensional collapse in supervised learning primarily requires conditioning the second moment (Covariance). This work proposes **Weak SIGReg**, which targets the covariance matrix directly using Randomized Numerical Linear Algebra (8).

This formulation relates to VICReg (9) and Barlow Twins (10) but applies the constraint purely as an internal regularizer via a direct Frobenius norm on the sketched covariance. This "sketching" step is crucial: it allows us to regularize high-dimensional layers (e.g., $C = 1024$) by computing the covariance on a small random projection (e.g., $K = 64$), significantly reducing memory cost $O(C^2) \to O(CK)$.

```python
def sigreg_weak_loss(x, sketch_dim=64):
    """
    Forces Covariance(x) ~ Identity via Sketching.
    Approximates the 2nd Moment constraint efficiently.
    """
    N, C = x.size()
    # 1. Sketching (Dimensionality Reduction)
    if C > sketch_dim:
        # Random projection preserves geometric structure (Johnson-Lindenstrauss)
        S = torch.randn(sketch_dim, C, device=x.device) / (C ** 0.5)
        x = x @ S.T
    else:
        sketch_dim = C

    # 2. Centering & Covariance
    x = x - x.mean(dim=0, keepdim=True)
    cov = (x.T @ x) / (N - 1 + 1e-6)

    # 3. Target Identity & Loss
    target = torch.eye(sketch_dim, device=x.device)

    # Minimize Frobenius norm distance to Identity
    return torch.norm(cov - target, p='fro')
```

Figure 3: Weak SIGReg Implementation

## 3 EXPERIMENTS

This work validates SIGReg on CIFAR-100, specifically targeting "pathological" setups where standard optimization fails. To ensure a fair comparison, all experiments (baseline and SIGReg) utilize gradient clipping (norm=1.0).

### 3.1 RESCUING VISION TRANSFORMERS (VIT)

Standard ViTs were trained using AdamW with aggressive augmentation (Mixup, CutMix, RandAugment). This regime is known to be unstable for ViTs without careful tuning.

**The Collapse:** As shown in Table 1, the baseline ViT optimization collapses, reaching only 20.73% accuracy. This indicates the model likely converged to a degenerate solution.

**The Fix:** Adding SIGReg completely stabilizes the training. The proposed **Weak SIGReg** recovers performance to **72.02%**, matching the performance of the computationally more expensive Strong SIGReg.

Table 1: ViT Optimization Stability (CIFAR-100). The baseline collapses while SIGReg converges.

| Optimizer | SIGReg | Top-1 Acc | Status |
|---|---|---|---|
| AdamW | None | 20.73% | **Collapse** |
| AdamW | **Strong (LeJEPA)** | 70.20% | Converged |
| AdamW | **Weak (Ours)** | **72.02%** | **Converged** |

### 3.2 COMPARISON WITH EXPERT TUNING

A common counter-argument is that ViTs can train if hyperparameters are tuned perfectly. The ViT baseline was manually optimized using specific weight decay, initialization, absolute positional embeddings, and LR schedules ($\eta_{min} = 1e - 5$).

Table 2 shows that while expert tuning recovers performance (70.76%), **Weak SIGReg (71.65%-72.71%)** matches or exceeds this performance *without* the need for such granular tuning. This suggests SIGReg acts as a robust default stabilizer.

Table 2: ViT Performance: Expert Tuning vs. SIGReg.

| Model Setup | SIGReg | Top-1 Acc |
|---|---|---|
| Expert-Tuned Baseline | None | 70.76% |
| Expert-Tuned Baseline | **Strong** | **72.71%** |
| Expert-Tuned Baseline | Weak | 71.65% |

### 3.3 VANILLA MLP STRESS TEST

To test gradient propagation limits, a **6-layer Vanilla MLP** (ReLU, No BatchNorm, No Residuals) is trained with pure SGD. In this setting, gradients often vanish or explode due to poor conditioning. Table 3 shows that Weak SIGReg increases accuracy from 26.77% to 42.17%. By forcing the covariance towards identity, SIGReg effectively acts as a "Soft Batch Normalization," maintaining well-conditioned gradients through deep linear layers.

Table 3: Vanilla MLP (6-Layers, Pure SGD, No BN). SIGReg significantly improves gradient flow.

| Augmentation | SIGReg | Top-1 Acc |
|---|---|---|
| None | None | 26.77% |
| None | Strong | 35.99% |
| None | **Weak** | **42.17%** |

## 4 CONCLUSION

This work presents an adaptation of LeJEPA's SIGReg for supervised learning stability. By deriving a simpler **Weak-SIGReg** formulation based on sketched covariance matching, it is demonstrated that geometric regularization is a powerful tool for optimization. It effectively rescues ViT training from collapse and enables deep MLPs to train without normalization layers, offering a mathematically grounded alternative to architectural heuristics.

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

## A APPENDIX

### A.1 IMPLEMENTATION DETAILS

All experiments were conducted on CIFAR-100. For Strong SIGReg, the implementation described in LeJEPA (7) was utilized with a sketch dimension of 64 and 17 integration points. For Weak SIGReg, a sketch dimension of 64 was used. The regularization strength $\alpha$ was set to 0.1 for all experiments unless otherwise noted.

### A.2 EXTENDED ABLATION STUDIES

### A.2.1 FULL RESULTS ON CIFAR-100

A comprehensive breakdown is presented across ResNet, ViT, and MLP architectures. This work compares pure SGD/AdamW baselines against Strong and Weak SIGReg.

Table 4: Complete Experimental Results across all architectures.

| Model | Optimizer | Augmentation | SIGReg | Epochs | Top-1 Acc |
|-------|-----------|--------------|--------|--------|-----------|
| ResNet18 | SGD | No | None | 1600 | 79.03% |
| ResNet18 | SGD | No | Strong | 1600 | 78.86% |
| ResNet18 | SGD | No | **Weak** | 1600 | **79.42%** |
| ResNet18 | SGD | Yes | None | 400 | 82.13% |
| ResNet18 | SGD | Yes | Strong | 400 | 81.18% |
| ResNet18 | SGD | Yes | **Weak** | 400 | **82.13%** |
| ViT | AdamW | Yes | None | 400 | 20.73% |
| ViT | AdamW | Yes | Strong | 400 | 70.20% |
| ViT | AdamW | Yes | **Weak** | 400 | **72.02%** |

Note: ResNet18 experiments show that SIGReg does not degrade performance on architectures that are already stable (via Batch Norm and Residuals), confirming it is safe to use as a default regularizer.

### A.2.2 FIXING THE ViT BASELINE

To interpret the failure of the ViT baseline (20.73%) in Table 1, an extensive manual intervention study was performed. This work applies a suite of heuristics commonly used to stabilize ViTs, including: specific weight decay, absolute position embeddings, specific initialization, LR schedulers ($\eta_{min} = 1e - 5$), and modified drop path rates.

Table 5: ViT Performance after Manual Intervention. SIGReg provides additive gains even after heavy manual tuning.

| Model | Optimizer | Augmentation | SIGReg | Top-1 Acc |
|-------|-----------|--------------|--------|-----------|
| ViT (Fixed) | AdamW | Yes | None | 70.76% |
| ViT (Fixed) | AdamW | Yes | **Strong** | **72.71%** |
| ViT (Fixed) | AdamW | Yes | Weak | 71.65% |

**Findings:**

1. **SIGReg vs. Heuristics:** In Table 1, Weak SIGReg achieved **72.02%** without any manual tuning. Table 5 shows that applying extensive manual heuristics ("ViT Fixed") achieves **70.76%**. This suggests SIGReg can effectively replace the need for complex, expert-level hyperparameter tuning.

2. **Additive Gains:** When SIGReg is added to the manually fixed ViT (Table 5), further gains was seen (up to 72.71%), indicating that geometric regularization provides benefits orthogonal to standard training heuristics.

A.2.3   ANALYSIS WITH MUON OPTIMIZER

To investigate whether SIGReg provides value beyond standard optimization choices, **Muon** was tested, a recent optimizer designed to improve stability via orthogonal updates. Muon was evaluated on the Vision Transformer setup where standard AdamW failed.

Table 6: ViT Performance with Muon Optimizer. Muon alone stabilizes the baseline significantly compared to AdamW (Table 1), yet SIGReg still provides substantial additive gains.

| Model Version | Augmentation | SIGReg | Top-1 Acc | Gain over Baseline |
|---|---|---|---|---|
| Standard ViT | No | None | 58.77% | - |
| Standard ViT | No | Strong | 63.16% | +4.39% |
| Standard ViT | No | **Weak** | **67.52%** | **+8.75%** |
| Standard ViT | Yes | None | 62.44% | - |
| Standard ViT | Yes | Strong | 74.34% | +11.90% |
| Standard ViT | Yes | **Weak** | **74.56%** | **+12.12%** |
| Fixed ViT | Yes | None | 75.87% | - |
| Fixed ViT | Yes | **Strong** | **76.98%** | **+1.11%** |
| Fixed ViT | Yes | Weak | 76.24% | +0.37% |

**Analysis:**

1. **Muon Stabilizes, SIGReg Optimizes:** While AdamW caused the Standard ViT to collapse (20.73%), Muon recovers it to a workable baseline (62.44%). However, adding Weak SIGReg further boosts this to **74.56%**. This demonstrates that SIGReg (which constrains representation geometry) and Muon (which constrains update geometry) are complementary.

2. **Peak Performance:** The combination of the "Fixed" ViT architecture, Muon optimizer, and Strong SIGReg yields the highest reported accuracy in this study (**76.98%**), suggesting that SIGReg remains relevant even in highly optimized, state-of-the-art training recipes.

A.2.4   VANILLA MLP STRESS TEST DETAILS

A 6-layer MLP was tested to isolate the effect of SIGReg on gradient propagation. The setup included: 1024 hidden dimension, ReLU activation, **No Dropout, No Batch Norm, No Residual connections**, and Pure SGD (no momentum).

Table 7: Vanilla MLP Optimization.

| Model | Optimizer | Augmentation | SIGReg | Top-1 Acc |
|---|---|---|---|---|
| MLP | SGD | No | None | 26.77% |
| MLP | SGD | No | Strong | 35.99% |
| MLP | SGD | No | **Weak** | **42.17%** |
| MLP | SGD | Yes | None | 38.08% |
| MLP | SGD | Yes | Strong | 38.70% |
| MLP | SGD | Yes | **Weak** | 38.40% |

It is hypothesized that the CutMix/MixUp version performed worse for Weak SIGReg (38.40% vs 42.17%) because the training schedule (400 epochs) was insufficient for the increased difficulty of the augmented task, whereas the un-augmented version converged faster.

