# OpenReview forum: "Weak-SIGReg: Covariance Regularization for Stable Deep Learning"
_ICLR.cc/2026/Workshop/GRaM — ICLR 2026 Workshop GRaM Poster_

### Official Review · Reviewer_R4Bp · 2026-02-20
**Review for Submission 37: Good intuition and good direction**

**Rating:** 6
**Confidence:** 5

**Review:**

Overview: The paper uses SIGReg (Sketched Isotropic Gaussian Regularization) for supervised training as an optimization stabilizer. The authors introduce Weak-SIGReg, a computationally efficient variant that enforces covariance isotropy via random sketching.

Strengths:
1. Demonstrates improved CIFAR-100 training on ViT from (20.73% to 72.02% accuracy) without any architectural changes.
2. SIGReg Computation via random sketching is a quite useful and fast technique to use in first/quasi-first order optimisers like SGD and AdamW.
3. Extensive experiments, but only training ViT for CIFAR100.

Weakness:
1. As this paper proposes an explicit regularization to loss function, keeping the problem setup (classification, loss function), and the regularized loss function would be beneficial for readers from varied communities.
2. Although the authors have shown good improvements in image classification, the reviewer believes that there is a substantial impact of this idea on learning autoregressive models (e.g. transformers) [1]. So it would be fair to show at least one table of how well this regularizer performs in a small pre-training / fine-tuning task.

References:
[1] Zhang, Yushun, et al. "Why Transformers need Adam: A Hessian Perspective." Advances in neural information processing systems 37 (2024): 131786-131823.

Relevance to topics listed in GRaM call for papers: Yes

Originality and novelty: Yes

Technical soundness of method: Yes

Clarity in writing and organization of the paper: Good experiments, but needs some theory preliminaries to show the overall picture.

For the Proceedings track: N/A

Double-blind reviewing: No violations of anonymity were found.

Use of LLMs: I feel that a few parts may have been generated by LLMs: (1)Section 3.3 Vannilla MLP Stress Test (There is no stress test here, its simple classification task, I have seen ChatGPT genereally tries to write words like that.) (2)A 2.3 Analysis Muon Stabilizes, SIGReg Optimizes - Please correct this line, Muon uses covariance square root and any optimizer which inverts the covaariance will not show substantial improvement as they already using a 2nd order optimizer.

**Pmlr Suitability:**

NA

---

### Official Review · Reviewer_1gKA · 2026-02-21
**Adaptation of SIGReg from LeJEPA to deep learning stability improvement as proposed can be useful**

**Rating:** 6
**Confidence:** 4

**Review:**

The paper utilizes Sketched Isotropic Gaussian Regularization (SIGReg), as used in the LeJEPA self-supervised framework, and uses it as
a general optimization stabilizer for supervised learning.

Overall, the method shows promise in reducing the training collapse without expert fine-tuning as shown by the experimental results section.

Some suggestions to improve the paper:

1) Provide a mathematical representation of the weak-SIGReg in Section 2.2
2) In the fundamental portion of LeJEPA structure, provide a mathematical description of the Empirical Characteristic Function (ECF) of the embeddings.
3) Is the method specific to vision transformers or can this be generalized? Readers would help from a discussion about that.
4) Is there any connection of this approach and flow-based methods that try to learn the underlying complex distribution starting from Gaussian?

**Pmlr Suitability:**

NA

---

### Official Review · Reviewer_htWU · 2026-02-24
**weak-sigreg: okay results on cifar, but limited scope**

**Rating:** 6
**Confidence:** 4

**Review:**

**Summary:**

This paper pulls the SIGReg idea from LeJEPA and tries to use it as a plug-and-play stabilizer for supervised learning. The "Weak" version specifically targets the covariance matrix using random sketches to keep the math at $O(CK)$. They show it can stop ViTs from collapsing on CIFAR-100 and help deep MLPs train without the usual normalization layers.

**Pros:**

* It actually fixes the ViT collapse issue, jumping from 20.73% to 72.02% accuracy on CIFAR-100 without needing the usual architectural "hacks".
* The computational cost is low enough to be practical for wide layers.
* I liked the Muon optimizer tests in the appendix; it shows the method is additive to modern update-geometry constraints.

**Cons:**

* The experiments are basically just CIFAR-100. It's a bit of a toy setup and I'm not convinced this $O(CK)$ approximation holds up on something like ImageNet-1k.
* There is a lack of comparison with standard "No-BatchNorm" baselines like NFNet or Fixup.
* Performance on the 6-layer MLP is pretty bad (42.17%), which makes me think the identity constraint might be killing the model's expressivity.
* The implementation details for the sketching are vague. Does the matrix $S$ get resampled every step? If it stays fixed, the model could easily learn to bypass the constraint in the null space of the projection.

**Workshop fit:** The paper is fine for a workshop. It’s a decent derivation of a known SSL tool for supervised stability, but the empirical part is too thin for a full conference track.

**Pmlr Suitability:**

Yes

---

### Official Review · Reviewer_bUa4 · 2026-02-25
**variant of SIGReg for more stable training**

**Rating:** 7
**Confidence:** 4

**Review:**

This paper proposes a variant of SIGReg from LeJEPA. Instead moment matching with the gaussian characteristic function, the authors propose to enforce that a random sketch of  emprical covariance matrix should be the identity matrix. This leads to a very simple algorithm that surprisingly seems to have a positive impact on training stability.

While the experiments on CIFAR-100 are convincing, a more comprehensive experimental setting would have strengthened the paper. Also, I think the template used for the paper is incorrect. It is only 4 pages, so it should be in the Tiny Paper Track not the Proceedings Track.

**Pmlr Suitability:**

No

---

### Meta-Review · Area_Chair_Mja8 · 2026-02-27

**Decision:**

Accept

**Metareview:**

Reviewers were impressed by the effectiveness of SIGReg in supervised learning.  They felt the experiments should be expanded for a full  proceedings paper.

**Other Comments:**

Submitted as long paper.  Accept as tiny paper.

**Relevance To Proceedings:**

Tiny paper — does not apply

**Relevance To Workshop:**

Yes — suitable for GRaM

---

### Decision · Program_Chairs · 2026-03-02

**Decision:**

Accept (Poster)

**Comment:**

Accepted as Tiny Paper (not proceedings paper).